# Graph Geometry Interaction Learning

**Shichao Zhu**[1,3], **Shirui Pan**[4], **Chuan Zhou**[2,3], **Jia Wu**[5], **Yanan Cao**[1,3]*, **Bin Wang**[6]

[1]Institute of Information Engineering, Chinese Academy of Sciences, Beijing, China
[2]Academy of Mathematics and Systems Science, Chinese Academy of Sciences, Beijing, China
[3]School of Cyber Security, University of Chinese Academy of Sciences, Beijing, China
[4]Faculty of Information Technology, Monash University, Melbourne, Australia
[5]Faculty of Science and Engineering, Macquarie University, Sydney, Australia
[6]Xiaomi AI Lab, Beijing, China
zhushichao@iie.ac.cn, shirui.pan@monash.edu, zhouchuan@amss.ac.cn,
jia.wu@mq.edu.au, caoyanan@iie.ac.cn, wangbin11@xiaomi.com

## Abstract

While numerous approaches have been developed to embed graphs into either Euclidean or hyperbolic spaces, they do not fully utilize the information available in graphs, or lack the flexibility to model intrinsic complex graph geometry. To utilize the strength of both Euclidean and hyperbolic geometries, we develop a novel Geometry Interaction Learning (GIL) method for graphs, a well-suited and efficient alternative for learning abundant geometric properties in graph. GIL captures a more informative internal structural features with low dimensions while maintaining conformal invariance of each space. Furthermore, our method endows each node the freedom to determine the importance of each geometry space via a flexible dual feature interaction learning and probability assembling mechanism. Promising experimental results are presented for five benchmark datasets on node classification and link prediction tasks.

## 1 Introduction

Learning from graph-structured data is an important task in machine learning, for which graph neural networks (GNNs) have shown unprecedented performance [1]. It is common for GNNs to embed data in the Euclidean space [2, 3, 4], due to its intuition-friendly generalization with powerful simplicity and efficiency. The Euclidean space has closed-form formulas for basic operations, and the space satisfies the commutative and associative laws for basic operations such as addition. According to the associated vector space operations and invariance to node and edge permutations required in GNNs [5], it is sufficient to operate directly in this space and feed the Euclidean embeddings as input to GNNs, which has led to impressive performance on standard tasks, including node classification, link prediction and etc. A surge of GNN models, from both spectral perspective [2, 6] or spatial perspective [3, 5, 7], as well as generalization of them for inductive learning for unseen nodes or other applications [4, 8, 9, 10], have been emerged recently.

Many real-word complex graphs exhibit a non-Euclidean geometry such as scale-free or hierarchical structure [11, 12, 13, 14]. Typical applications include Internet [15], social network [16] and biological network. In these cases, the Euclidean spaces suffer from large distortion embeddings, while hyperbolic spaces offer an efficient alternative to embed these graphs with low dimensions, where the nodes and distances increase exponentially with the depth of the tree [17]. HGNN [18] and HGCN [19] are the most recent methods to generalize hyperbolic neural networks [20, 21] to graph domains, by moving the aggregation operation to the *tangent* space, where the operations satisfy the

---

permutation invariance required in GNNs. Due to the hierarchical modeling capability of hyperbolic geometry, these methods outperform Euclidean methods on scale-free graphs in standard tasks. As another line of works, the mixed-curvature product attempted to embed data into a product space of spherical, Euclidean and hyperbolic manifold, providing a unified embedding paradigm of different geometries [22, 23].

The existing works assume that all the nodes in a graph share the same spatial curvature without interaction between different spaces. Different from these works, this paper attempts to embed a graph into a Euclidean space and a hyperbolic space simultaneously in a new manner of interaction learning, with the hope that the two sets of embedding can mutually enhance each other and provide more effective representation for downstream tasks. Our motivation stems from the fact that real-life graph structures are complicated and have diverse properties. As shown in Figure 1, part of the graph (blue nodes) shows regular structures which can be easily captured in a Euclidean space, another part of the graph (yellow nodes) exhibits hierarchical structures which are better modelled in a hyperbolic space. Learning graph embedding in a single space will unfortunately result in sub-optimal results due to its incapacity to capture the nodes showing different structure properties. One straightforward solution is to learn graph embeddings with different graph neural networks in different spaces and concatenate the embeddings into a single vector for downstream graph analysis tasks. However, as will be shown in our experiments in Sec 4.2, such an approach cannot produce a satisfactory result because both embeddings are learned separately and have been largely distorted in the learning process by the nodes which exhibit different graph structures.

In this paper we advocate a novel geometry interaction learning (GIL) method for graphs, which exploits hyperbolic and Euclidean geometries jointly for graph embedding through an interaction learning mechanism to exploit the reliable spatial features in graphs. GIL not only considers how to interact different geometric spaces, but also ensures the conformal invariance of each space. Furthermore, GIL takes advantages of the convenience of operations in Euclidean spaces and preserve the ability to model complex structure in hyperbolic spaces.

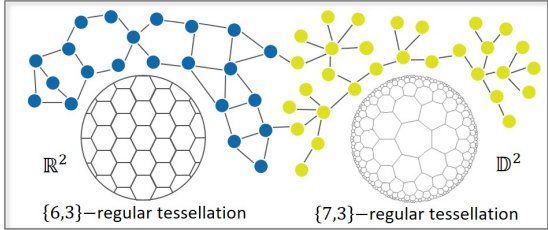

Figure 1: Example of a graph with both hyperbolic and Euclidean structures. Euclidean space (left) is suitable for representing the blue points with low-dimensional regular structure, while yellow points with hierarchical structure are more suitable for hyperbolic space (right). Here $\{n, k\}$-regular tessellation consists of $n$-edge regular polygons with $k$ edges meeting at each vertex.

In the new approach, we first utilize the *exponential* and *logarithmic* mappings as bridges between two spaces for geometry message propagation. A novel distance-aware attention mechanism for hyperbolic message propagation, which approximates the neighbor information on the *tangent* space guided by hyperbolic distance, is designed to capture the information in hyperbolic spaces. After the message propagation in local space, node features are integrated via an interaction learning module adaptively from dual spaces to adjust the conformal structural features. In the end, we perform hyperplane-based logistic regression and probability assembling to obtain the category probabilities customized for each node, where the assembling weight determines the importance of each embedding from different geometry for downstream tasks.

Our extensive evaluations demonstrate that our method outperforms the SOTA methods in node classification and link prediction tasks on five benchmark datasets. The major contributions of this paper are summarized as follows.

- To our best knowledge, we are the first to exploit graph embedding in both Euclidean spaces and hyperbolic spaces simultaneously to better adapt for complex graph structure.

- We present a novel end-to-end method Geometry Interaction Learning (GIL) for graphs. Through dual feature interaction and probability assembling, our GIL method learns reliable spatial structure features in an adaptive way to maintain the conformal invariance.

- Extensive experimental results have verified the effectiveness of our method for node classification and link prediction on benchmark datasets.

## 2 Notation and Preliminaries

In this section, we recall some backgrounds on hyperbolic geometry, as well as some elementary operations of graph attention networks.

**Hyperbolic Geometry.** A Riemannian manifold $(\mathcal{M}, g)$ of dimension $n$ is a real and smooth manifold equipped with an inner product on *tangent* space $g_{\boldsymbol{x}} : \mathcal{T}_{\boldsymbol{x}}\mathcal{M} \times \mathcal{T}_{\boldsymbol{x}}\mathcal{M} \to \mathbb{R}$ at each point $\boldsymbol{x} \in \mathcal{M}$, where the *tangent* space $\mathcal{T}_{\boldsymbol{x}}\mathcal{M}$ is a $n$-dimensional vector space and can be seen as a first order local approximation of $\mathcal{M}$ around point $\boldsymbol{x}$. In particular, hyperbolic space $(\mathbb{D}_c^n, g^c)$, a constant negative curvature Riemannian manifold, is defined by the manifold $\mathbb{D}_c^n = \{\boldsymbol{x} \in \mathbb{R}^n : c\|\boldsymbol{x}\| < 1\}$ equipped with the following Riemannian metric: $g_{\boldsymbol{x}}^c = \lambda_{\boldsymbol{x}}^2 g^E$, where $\lambda_{\boldsymbol{x}} := \frac{2}{1-c\|\boldsymbol{x}\|^2}$ and $g^E = \boldsymbol{I}_n$ is the Euclidean metric tensor.

Here, we adopt Poincaré ball model, which is a compact representation of hyperbolic space and has the principled generalizations of basic operations (e.g. addition, multiplication), as well as closed-form expressions for basic objects (e.g. distance) derived by [20]. The connections between hyperbolic space and *tangent* space are established by the *exponential* map $\exp_{\boldsymbol{x}}^c : \mathcal{T}_{\boldsymbol{x}}\mathbb{D}_c^n \to \mathbb{D}_c^n$ and the *logarithmic* map $\log_{\boldsymbol{x}}^c : \mathbb{D}_c^n \to \mathcal{T}_{\boldsymbol{x}}\mathbb{D}_c^n$ as follows.

$$\exp_{\boldsymbol{x}}^c(\boldsymbol{v}) = \boldsymbol{x} \oplus_c \left( \tanh\left( \sqrt{c}\frac{\lambda_{\boldsymbol{x}}^c\|\boldsymbol{v}\|}{2} \right) \frac{\boldsymbol{v}}{\sqrt{c}\|\boldsymbol{v}\|} \right), \tag{1}$$

$$\log_{\boldsymbol{x}}^c(\boldsymbol{y}) = \frac{2}{\sqrt{c}\lambda_{\boldsymbol{x}}^c} \tanh^{-1}\left( \sqrt{c}\|-\boldsymbol{x} \oplus_c \boldsymbol{y}\| \right) \frac{-\boldsymbol{x} \oplus_c \boldsymbol{y}}{\|-\boldsymbol{x} \oplus_c \boldsymbol{y}\|}, \tag{2}$$

where $\boldsymbol{x}, \boldsymbol{y} \in \mathbb{D}_c^n, \boldsymbol{v} \in \mathcal{T}_{\boldsymbol{x}}\mathbb{D}_c^n$. And the $\oplus_c$ represents *Möbius addition* as follows.

$$\boldsymbol{x} \oplus_c \boldsymbol{y} := \frac{(1 + 2c\langle\boldsymbol{x}, \boldsymbol{y}\rangle + c\|\boldsymbol{y}\|^2)\boldsymbol{x} + (1 - c\|\boldsymbol{x}\|^2)\boldsymbol{y}}{1 + 2c\langle\boldsymbol{x}, \boldsymbol{y}\rangle + c^2\|\boldsymbol{x}\|^2\|\boldsymbol{y}\|^2}. \tag{3}$$

Similarly, the generalization for multiplication in hyperbolic space can be defined by the *Möbius scalar multiplication* and *Möbius matrix multiplication* between vector $\boldsymbol{x} \in \mathbb{D}_c^n \setminus \{\boldsymbol{0}\}$.

$$r \otimes_c \boldsymbol{x} := \frac{1}{\sqrt{c}}\tanh\left( r\tanh^{-1}\left( \sqrt{c}\|\boldsymbol{x}\| \right) \right) \frac{\boldsymbol{x}}{\|\boldsymbol{x}\|}, \tag{4}$$

$$M \otimes_c \boldsymbol{x} := (1/\sqrt{c})\tanh\left( \frac{\|M\boldsymbol{x}\|}{\|\boldsymbol{x}\|}\tanh^{-1}(\sqrt{c}\|\boldsymbol{x}\|) \right) \frac{M\boldsymbol{x}}{\|M\boldsymbol{x}\|}, \tag{5}$$

where $r \in \mathbb{R}$ and $M \in \mathbb{R}^{m \times n}$.

**Graph Attention Networks.** Graph Attention Networks (GAT) [24] can be interpreted as performing attentional message propagation and updating on nodes. At each layer, the node feature $h_i \in \mathbb{R}^F$ will be updated by its attentional neighbor messages to $h_i' \in \mathbb{R}^{F'}$ as follows.

$$h_i' = \sigma\left( \sum_{j \in \{i\} \cup \mathcal{N}_i} \alpha_{ij} W h_j \right), \tag{6}$$

where $\sigma$ is a non-linear activation, $\mathcal{N}_i$ denotes the one-hop neighbors of node $i$, and $W \in \mathbb{R}^{F' \times F}$ is a trainable parameter. The attention coefficient $\alpha_{ij}$ is computed as follows, parametrized by a weight vector $\boldsymbol{a} \in \mathbb{R}^{2F'}$.

$$e_{ij} = \text{LeakyRelu}\left( \boldsymbol{a}^\top[W h_i \| W h_j] \right), \ \alpha_{ij} = \text{softmax}_j(e_{ij}) = \frac{\exp(e_{ij})}{\sum_{k \in \mathcal{N}_i} \exp(e_{ik})}. \tag{7}$$

## 3 Graph Geometry Interaction Learning

Our approach exploits different geometry spaces through a feature interaction scheme, based on which a probability assembling is further developed to integrate the category probabilities for the final prediction. Figure 2 shows our conceptual framework. We detail our method below.

### 3.1 Geometry Feature Interaction

**Geometry Message Propagation.** GNNs essentially employ message propagation to learn the node embedding. For message propagation in Euclidean spaces, we can apply a GAT to propagate message

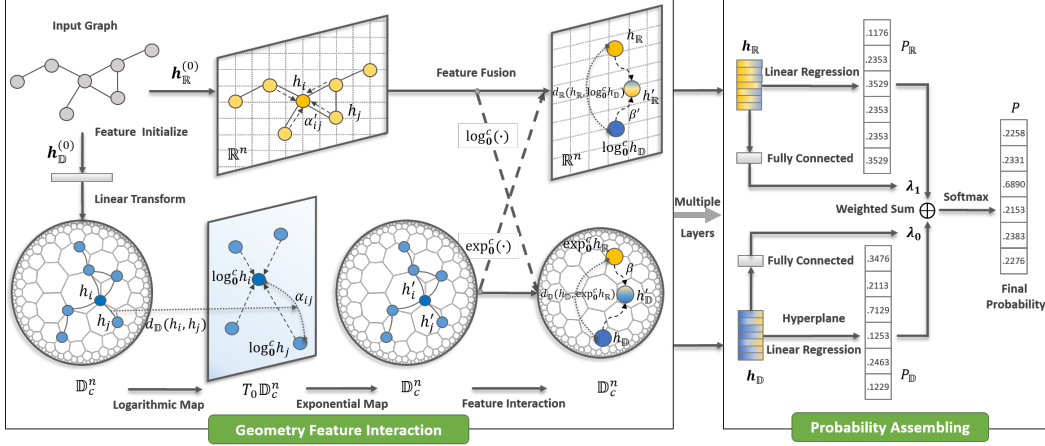

Figure 2: Schematic of GIL architecture. (i) Geometry Message Propagation: propagates neighbor information simultaneously in a Euclidean space and a hyperbolic space via a distance-aware attention mechanism. In hyperbolic space, messages are aggregated through a distance-guided self-attention mechanism on the *tangent* space; (ii) Dual Feature Interaction: node features undergo an adaptive adjustment from the dual feature space based on distance similarity; (iii) Probability Assembling: the weighted sum of the classification probability from each space is used to obtain the final classification result during the training phase. The method is trained end-to-end.

for each node, as defined in Eq. (6). However, it is not sufficient to operate directly in hyperbolic space to maintain the permutation invariance, as the basic operations such as *Möbius addition* cannot hold the property of commutative and associative, like $x \oplus_c y \neq y \oplus_c x$. The general efficient approach is to move basic operations to the *tangent* space, used in the recent works [18, 19]. This operation is also intrinsic, which has been proved and called *tangential aggregations* in [22].

Based on above settings, we develop the distance-aware attention message propagation for hyperbolic space, leveraging the hyperbolic distance self-attentional mechanism to address the information loss of prior methods based on graph convolutional approximations on *tangent* space [18, 19]. In particular, given a graph and nodes features, we first apply $\exp_0^c(\cdot)$ to map Euclidean input features into hyperbolic space and then initialize $h^{(0)}$. After a linear transformation of node features in hyperbolic space, we apply $\log_0^c(\cdot)$ to map nodes into *tangent* space to prepare for subsequent message propagation. To preserve more information of hyperbolic geometry, we aggregate neighbor messages through performing self-attention guided by hyperbolic distance between nodes. After aggregating messages, we apply a pointwise non-linearity $\sigma(\cdot)$, typically ReLU. Finally, map node features back to hyperbolic space. Specifically, for every propagation layer $k \in \{0, 1, ..., K-1\}$, the operations are given as follows.

$$m_i^{(k+1)} = \sum_{j \in \{i\} \cup \mathcal{N}_i} \alpha_{ij}^{(k)} \log_0^c \left( W^{(k)} \otimes_c h_j^{(k)} \oplus_c b^{(k)} \right), \tag{8}$$

$$h_i^{(k+1)} = \exp_0^c \left( \sigma \left( m_i^{(k+1)} \right) \right), \tag{9}$$

where $h_j^{(k)} \in \mathbb{D}_c^{F^{(k)}}$ denotes the hidden features of node $j$, $\otimes_c$ and $\oplus_c$ denote the *Möbius matrix multiplication* and *addition*, respectively. $W^{(k)} \in \mathbb{R}^{F^{(k+1)} \times F^{(k)}}$ is a trainable weight matrix, and $b^{(k)} \in \mathbb{D}_c^{F^{(k+1)}}$ denotes the bias. After mapping into *tangent* space, messages are scaled by the normalized attention $\alpha_{ij}^{(k)}$ of node $j$'s features to node $i$, computed as below Eq. (10). And $h_j^{(k+1)} \in \mathbb{D}_c^{F^{(k+1)}}$ will be updated by the messages $m_i^{(k+1)} \in \mathbb{R}^{F^{(k+1)}}$ aggregated from neighbor nodes.

*Distance-aware attention.* This part devotes to how to calculate $\alpha_{ij}^{(k)}$. For the sake of convenience, we omit all superscripts $(k)$ in this part. For example, $\alpha_{ij}^{(k)}$ can be reduced to $\alpha_{ij}$. For the distance guided self-attention on the node features, there is a shared attentional mechanism: $\mathbb{R}^{2F} \times \mathbb{R} \to \mathbb{R}$ to

compute attention coefficients as follows.

$$e_{ij} = \text{LeakyRelu}\left(\boldsymbol{a}^\top[\hat{h}_i||\hat{h}_j] \times d_{\mathbb{D}_c}(h_i, h_j)\right), \ \alpha_{ij} = \text{softmax}_j(e_{ij}) = \frac{\exp(e_{ij})}{\sum_{k \in \mathcal{N}_i} \exp(e_{ik})}, \quad (10)$$

$$d_{\mathbb{D}_c}(h_i, h_j) = (2/\sqrt{c})\tanh^{-1}(\sqrt{c}|| - h_i \oplus_c h_j||), \quad (11)$$

where $||$ denotes the concatenation operation, $\boldsymbol{a} \in \mathbb{R}^{2F}$ is a trainable vector, and $\hat{h}_j = \log_{\boldsymbol{0}}^c(W \otimes_c h_j)$ denotes the nodes features on *tangent* space, $e_{ij}$ indicates the importance of node $j$'s features on *tangent* space to node $i$. Take the hyperbolic relationship of nodes into account, we introduce the normalized hyperbolic distance $d_{\mathbb{D}_c}(\cdot, \cdot)$ between nodes to adapt for the attention, computed as Eq. (11). To make coefficients easily comparable across different nodes, we normalize them across all choices of $j$ using the softmax function.

**Dual Feature Interaction.** After propagating messages based on graph topology, node embeddings will undergo an adaptive adjustment from dual feature spaces to promote interaction between different geometries. Specifically, the hyperbolic embeddings $\boldsymbol{h}_{\mathbb{D}_c}^{(k+1)} \in \mathbb{D}_c^{N \times F^{(k+1)}}$ and Euclidean embeddings $\boldsymbol{h}_{\mathbb{R}}^{(k+1)} \in \mathbb{R}^{N \times F^{(k+1)}}$ will be adjusted based on the similarity of two embeddings. In order to set up the interaction between different geometries, we apply $\exp_{\boldsymbol{0}}^c(\cdot)$ to map Euclidean embeddings into hyperbolic space, and reversely, apply $\log_{\boldsymbol{0}}^c(\cdot)$ to map hyperbolic embeddings into the Euclidean space. To simplify the formula, $\boldsymbol{h}_{\mathbb{D}_c}^{(k+1)}$ and $\boldsymbol{h}_{\mathbb{R}}^{(k+1)}$ are represented by $\boldsymbol{h}_{\mathbb{D}_c}$ and $\boldsymbol{h}_{\mathbb{R}}$.

$$\boldsymbol{h}'_{\mathbb{D}_c} = \boldsymbol{h}_{\mathbb{D}_c} \oplus_c \left(\beta d_{\mathbb{D}_c}\left(\boldsymbol{h}_{\mathbb{D}_c}, \exp_0^c(\boldsymbol{h}_{\mathbb{R}})\right) \otimes_c \exp_0^c(\boldsymbol{h}_{\mathbb{R}})\right), \quad (12)$$

$$\boldsymbol{h}'_{\mathbb{R}} = \boldsymbol{h}_{\mathbb{R}} + \left(\beta' d_{\mathbb{R}}\left(\boldsymbol{h}_{\mathbb{R}}, \log_0^c(\boldsymbol{h}_{\mathbb{D}_c})\right) \times \log_0^c(\boldsymbol{h}_{\mathbb{D}_c})\right), \quad (13)$$

where $\boldsymbol{h}'_{\mathbb{D}_c}$ and $\boldsymbol{h}'_{\mathbb{R}}$ denote the fused node embeddings, $\oplus_c$ and $\otimes_c$ are the *möbius addition* and *möbius scalar multiplication*, respectively. The similarity from different spatial features is measured by hyperbolic distance $d_{\mathbb{D}_c}(\cdot, \cdot)$ as Eq. (11) and Euclidean distance $d_{\mathbb{R}}(\cdot, \cdot)$, parameterized by two trainable weights $\beta, \beta' \in \mathbb{R}$. After interactions for $K$ layers, we will get two fusion geometric embeddings for each node.

*Conformal invariance.* It is worth mentioning that after the feature fusion, the node embeddings not only integrate different geometric characteristics via interaction learning, but also maintain properties and structures of original space. That is the fused node embeddings $\boldsymbol{h}'_{\mathbb{D}_c} \in \mathbb{D}_c^{N \times F^{(k+1)}}$ and $\boldsymbol{h}'_{\mathbb{R}} \in \mathbb{R}^{N \times F^{(k+1)}}$ are still located at their original space, which satisfy their corresponding Riemannian metric $g_{\boldsymbol{x}}^c = \lambda_{\boldsymbol{x}}^2 g^E$ at each point $\boldsymbol{x}$, maintaining the same conformal factor $\lambda$, respectively.

## 3.2 Hyperplane-Based Logistic Regression and Probability Assembling

**Hyperplane-Based Logistic Regression.** In order to perform multi-class logistic regression on the hyperbolic space, we generalize Euclidean softmax to hyperbolic softmax by introducing the affine hyperplane as decision boundary. And the category probability is measured by the distance from node to hyperplane. In particular, the standard logistic regression in Euclidean spaces can be formulated as learning a hyperplane for each class [25], which can be expressed as follows.

$$p(y|\boldsymbol{x}; \boldsymbol{u}) \propto \exp(y\|\boldsymbol{u}\|\langle \boldsymbol{x} - \boldsymbol{p}, \boldsymbol{u}\rangle) \quad (14)$$

$$= \exp(y\,\text{sign}(\langle \boldsymbol{x} - \boldsymbol{p}, \boldsymbol{u}\rangle)\|\boldsymbol{u}\|d_{\mathbb{R}}(\boldsymbol{x}, H_{\boldsymbol{u},\boldsymbol{p}})), \quad (15)$$

where $\langle\cdot, \cdot\rangle$ and $\|\cdot\|$ denote the Euclidean inner product and norm, $y \in \{-1, 1\}$, and $d_{\mathbb{R}}$ is the Euclidean distance of $\boldsymbol{x}$ from the hyperplane $H_{\boldsymbol{u},\boldsymbol{p}} = \{\boldsymbol{x} \in \mathbb{R}^n : \langle \boldsymbol{x} - \boldsymbol{p}, \boldsymbol{u}\rangle = 0\}$ that corresponds to the normal vector $\boldsymbol{u}$ and a point $\boldsymbol{p}$ located at $H_{\boldsymbol{u},\boldsymbol{p}}$.

To define a hyperplane in hyperbolic space, we should figure out the normal vector, which completely determines the direction of hyperplane. Since the *tangent* space is Euclidean, the normal vector and inner product can be intuitively defined in this space. Therefore, we can define the hyperbolic affine hyperplane $\tilde{H}_{\boldsymbol{u},\boldsymbol{p}}^c$ with normal vector $\boldsymbol{u} \in \mathcal{T}_{\boldsymbol{p}}\mathbb{D}_c^n \backslash \{\boldsymbol{0}\}$ and point $\boldsymbol{p} \in \mathbb{D}_c^n$ located at the hyperplane as follows.

$$\tilde{H}_{\boldsymbol{u},\boldsymbol{p}}^c = \left\{\boldsymbol{x} \in \mathbb{D}_c^n \ : \ \langle\log_{\boldsymbol{p}}^c(\boldsymbol{x}), \boldsymbol{u}\rangle_{\boldsymbol{p}} = 0\right\} \quad (16)$$

$$= \{\boldsymbol{x} \in \mathbb{D}_c^n \ : \ \langle \boldsymbol{x} \ominus_c \boldsymbol{p}, \boldsymbol{u}\rangle = 0\} \quad (17)$$

where $\ominus_c$ denotes the *Möbius substraction* that can be represented via *Möbius addition* as $\boldsymbol{x} \ominus_c \boldsymbol{y} = \boldsymbol{x} \oplus_c (-\boldsymbol{y})$. For the distance between $\boldsymbol{x} \in \mathbb{D}_c^n$ and hyperplane $\tilde{H}_{\boldsymbol{u},\boldsymbol{p}}^c$ is defined by [20] as

$$d_{\mathbb{D}_c}(\boldsymbol{x}, \tilde{H}_{\boldsymbol{u},\boldsymbol{p}}^c) := \inf_{\boldsymbol{w} \in \tilde{H}_{\boldsymbol{u},\boldsymbol{p}}^c} d_{\mathbb{D}_c}(\boldsymbol{x}, \boldsymbol{w}) \tag{18}$$

$$= \frac{1}{\sqrt{c}} \sinh^{-1}\left( \frac{2\sqrt{c}|\langle -\boldsymbol{p} \oplus_c \boldsymbol{x}, \boldsymbol{u} \rangle|}{(1 - c\|-\boldsymbol{p} \oplus_c \boldsymbol{x}\|)\|\boldsymbol{u}\|} \right). \tag{19}$$

Based on above generalizations, we can derive the hyperbolic probability for each class in the form of Eq. (15).

**Probability Assembling.** Intuitively, it is natural to concatenate the different geometric features and then feed the single feature into one classifier. However, because of the different operations of geometric spaces, one of the conformal invariances has to be abandoned, either preserving the space form of Euclidean Eq. (20) or hyperbolic Eq. (21).

$$P = \text{Euc-Softmax}\left(f\left(\log_0^c(\boldsymbol{X}_{\mathbb{D}_c})\|\boldsymbol{X}_{\mathbb{R}}\right)\right), \tag{20}$$
$$P = \text{Hyp-Softmax}\left(g\left(\boldsymbol{X}_{\mathbb{D}_c}\|\exp_0^c(\boldsymbol{X}_{\mathbb{R}})\right)\right), \tag{21}$$

where Euc-Softmax and Hyp-Softmax are Euclidean softmax and Hyperbolic softmax, respectively. And function $f$ and $g$ are two linear mapping functions in respective geometric space.

Here, we want to maintain characteristics of each space at the same time, preserving as many spatial attributes as possible. Moreover, to recall the aforementioned problem, we try to figure out which geometric embeddings are more critical for each node and then assign more weight to the corresponding probability. Therefore, we design the probability assembling as follows.

$$P = \boldsymbol{\lambda}_0 P_{\mathbb{D}_c} + \boldsymbol{\lambda}_1 P_{\mathbb{R}}, \quad s.t. \ \boldsymbol{\lambda}_0 + \boldsymbol{\lambda}_1 = \boldsymbol{1}, \tag{22}$$

where $P_{\mathbb{D}_c}, P_{\mathbb{R}} \in \mathbb{R}^{N \times M}$ is the probability from hyperbolic and Euclidean spaces, respectively, $N$ denotes the number of nodes, and $M$ denotes the number of classes. The corresponding weights $\boldsymbol{\lambda}_0, \boldsymbol{\lambda}_1 \in \mathbb{R}^N$ denote the node-level weights for different geometric probabilities, and satisfy the normalization condition by performing $L_2$ normalization of concatenation $[\boldsymbol{\lambda}_0, \boldsymbol{\lambda}_1]$ along dimension 1. They are computed by two fully connected layers with input of respective node embeddings $\boldsymbol{X}_{\mathbb{D}_c} \in \mathbb{D}_c^{N \times F^{(K)}}$ and $\boldsymbol{X}_{\mathbb{R}} \in \mathbb{R}^{N \times F^{(K)}}$ as follows.

$$\boldsymbol{\lambda}_0 = \text{sigmoid}\left(FC_0\left(\log_0^c(\boldsymbol{X}_{\mathbb{D}_c}^\top)\right)\right), \ \boldsymbol{\lambda}_1 = \text{sigmoid}\left(FC_1\left(\boldsymbol{X}_{\mathbb{R}}^\top\right)\right),$$
$$[\boldsymbol{\lambda}_0, \boldsymbol{\lambda}_1] = \frac{[\boldsymbol{\lambda}_0, \boldsymbol{\lambda}_1]}{\max\left(\|[\boldsymbol{\lambda}_0, \boldsymbol{\lambda}_1]\|_2, \epsilon\right)}, \tag{23}$$

where $\epsilon$ is a small value to avoid division by zero. As we can see, the contribution of two spatial probabilities to the final probability is determined by nodes features in corresponding space. In other words, the nodes themselves determine which probability is more reliable for the downstream task.

For link prediction, we model the probability of an edge as proposed by [26] via the Fermi-Dirac distribution as follows,

$$P(e_{ij} \in \mathcal{E}|\Theta) = \frac{1}{e^{(d(\boldsymbol{x}_i, \boldsymbol{x}_j) - r)/t} + 1}, \tag{24}$$

where $r, t > 0$ are hyperparameters. The probability assembling for link prediction follows the same way of node classification. That is, by replacing node embeddings as the subtraction of two endpoints embeddings in Eq. (23), to learn corresponding weights for the probability of each edge in two spaces, and then view the weighted sum as the final probability.

### 3.3 Model Analysis

**Architecture.** We implement our method with an end-to-end architecture as shown in Figure 2. The left panel in the figure shows the message propagation and interaction between two spaces. For each layer, local messages diffuse across the whole graph, and then node features are adaptively merged to adjust themselves according to dual geometric information. After stacking several layers to make geometric features deeply interact, GIL applies the probability assembling to get the reliable final result as shown in the right panel. Note that care has to be taken to achieve numerical stability through applying safe projection. Our method is trained by minimizing the cross-entropy loss over all labeled examples for node classification, and for link prediction, by minimizing the cross-entropy loss using negative sampling.

**Complexity.** For message propagation module, the time complexity is $O(NFF' + |\mathcal{E}|F')$, where $N$ and $|\mathcal{E}|$ are the number of nodes and edges, $F$ and $F'$ are the dimension of input and hidden features, respectively. The operation of distance-aware self-attention can be parallelized across all edges. The computation of remaining parts can be parallelized across all nodes and it is computationally efficient.

## 4 Experiments

In this section, we conduct extensive experiments to verify the performance of GIL on node classification and link prediction. Code and data are available at `https://github.com/CheriseZhu/GIL`.

### 4.1 Experiments Setup

**Datasets.** For node classification and link prediction tasks, we consider five benchmark datasets:

Table 1: A summary of the benchmark datasets. For each dataset, we report the number of nodes, number of edges, number of classes, dimensions of input features, and hyperbolicity distribution.

| Dataset | Nodes | Edges | Classes | Features | hyperbolicity distribution | | | | | |
|---|---|---|---|---|---|---|---|---|---|---|
| | | | | | 0 | 0.5 | 1 | 1.5 | 2 | 2.5 |
| Disease | 1,044 | 1,043 | 2 | 1,000 | 1.0000 | 0 | 0 | 0 | 0 | 0 |
| Airport | 3,188 | 18,631 | 4 | 4 | 0.6376 | 0.3563 | 0.0061 | 0 | 0 | 0 |
| Pubmed | 19,717 | 44,338 | 3 | 500 | 0.4239 | 0.4549 | 0.1094 | 0.0112 | 0.0006 | 0 |
| Citeseer | 3,327 | 4,732 | 6 | 3,703 | 0.3659 | 0.3538 | 0.1699 | 0.0678 | 0.0288 | 0 |
| Cora | 2,708 | 5,429 | 7 | 1,433 | 0.4474 | 0.4073 | 0.1248 | 0.0189 | 0.0016 | 0.0102 |

Disease, Airport, Cora, Pubmed and Citeseer. Dataset statistics are summarized in Table 1. The first two datasets are derived by [19]. Disease dataset simulates the disease propagation tree, where node represents a state of being infected or not by SIR disease. Airport dataset describes the location of airports and airline networks, where nodes represent airports and edges represent airline routes. In the citation network datasets: Cora, Pubmed and Citeseer [27], where nodes are documents and edges are citation links. In addition, we compute the hyperbolicity distribution of the maximum connected subgraph in each datatset to characterize the degree of how tree-like a graph exists over the graph, defined by [28]. And the lower hyperbolicity is, the more hyperbolic the graph is. As shown in Table 1, the datasets exhibit multiple different hyperbolicities in a graph, except Disease dataset, which is a tree with 0-hyperbolic.

Table 2: Link prediction (LP) in ROC AUC and node classification (NC) in Accuracy with the std.

| Method | Disease | | Airport | | Pubmed | | Citeseer | | Cora | |
|---|---|---|---|---|---|---|---|---|---|---|
| | LP | NC | LP | NC | LP | NC | LP | NC | LP | NC |
| EUC | 69.41±1.41 | 32.56±1.19 | 92.00±0.00 | 60.90±3.40 | 79.82±2.87 | 48.20±0.76 | 80.15±0.86 | 61.28±0.91 | 84.92±1.17 | 23.80±0.80 |
| HYP | 73.00±1.43 | 45.52±3.09 | 94.57±0.00 | 70.29±0.40 | 84.91±0.17 | 68.51±0.37 | 79.65±0.86 | 61.71±0.74 | 86.64±0.61 | 22.13±0.97 |
| MLP | 63.65±2.63 | 28.80±2.23 | 89.81±0.56 | 68.90±0.46 | 83.33±0.56 | 72.40±0.21 | 93.73±0.63 | 59.53±0.90 | 83.33±0.56 | 51.59±1.28 |
| HNN | 71.26±1.96 | 41.18±1.85 | 90.81±0.23 | 80.59±0.46 | 94.69±0.06 | 69.88±0.43 | 93.30±0.52 | 59.50±1.28 | 90.92±0.40 | 54.76±0.61 |
| GCN | 58.00±1.41 | 69.79±0.54 | 89.31±0.43 | 81.59±0.61 | 89.56±3.66 | 78.10±0.43 | 82.56±1.92 | 70.35±0.41 | 90.47±0.24 | 81.50±0.53 |
| GAT | 58.16±0.92 | 70.40±0.49 | 90.85±0.23 | 81.59±0.36 | 91.46±1.82 | 78.21±0.44 | 86.48±1.50 | 71.58±0.80 | 93.17±0.20 | 83.03±0.50 |
| SAGE | 65.93±0.29 | 70.10±0.49 | 90.41±0.53 | 82.19±0.45 | 86.21±0.82 | 77.45±2.38 | 92.05±0.39 | 67.51±0.76 | 85.51±0.50 | 77.90±2.50 |
| SGC | 65.34±0.28 | 70.94±0.59 | 89.83±0.32 | 80.59±0.16 | 94.10±0.12 | 78.84±0.18 | 91.35±1.68 | 71.44±0.75 | 91.50±0.21 | 81.32±0.50 |
| HGNN | 81.54±1.22 | 81.27±3.53 | 96.42±0.44 | 84.71±0.98 | 92.75±0.26 | 77.13±0.82 | 93.58±0.33 | 69.99±1.00 | 91.67±0.41 | 78.26±1.19 |
| HGCN | 90.80±0.31 | 88.16±0.76 | 96.43±0.12 | 89.26±1.27 | 95.13±0.14 | 76.53±0.63 | 96.63±0.09 | 68.04±0.59 | 93.81±0.14 | 78.03±0.98 |
| HGAT | 87.63±1.67 | 90.30±0.62 | 97.86±0.09 | 89.62±1.03 | 94.18±0.18 | 77.42±0.66 | 95.84±0.37 | 68.64±0.30 | 94.02±0.18 | 78.32±1.39 |
| EucHyp | 97.91±0.32 | 90.13±0.24 | 96.12±0.26 | 90.71±0.89 | 94.12±0.25 | 78.40±0.65 | 94.62±0.14 | 68.89±1.24 | 93.63±0.10 | 81.38±0.62 |
| GIL | **99.90±0.31** | **90.70±0.40** | **98.77±0.30** | **91.33±0.78** | **95.49±0.16** | **78.91±0.26** | **99.85±0.45** | **72.97±0.67** | **98.28±0.85** | **83.64±0.63** |

**Baselines.** We compare our method to several state-of-the-art baselines, including 2 shallow methods, 2 NN-based methods, 4 Euclidean GNN-based methods and 3 hyperbolic GNN-based methods, to evaluate the effectiveness of our method. The details follow. *Shallow methods*: By utilizing the Euclidean and hyperbolic distance function as objective, we consider the Euclidean-based method (EUC) and Poincaré ball-based method (HYP) [29] as two shallow baselines. *NN methods*: Without regard to graph topology, MLP serves as the traditional feature-based deep learning method and HNN[20] serves as its variant in hyperbolic space. *GNNs methods*: Considering deep embeddings and graph structure simultaneously, GCN [2], GAT [24], SAGE [4] and SGC [30] serve as Euclidean GNN-based methods. HGNN [18], HGCN [19] and HGAT [21] serve as hyperbolic variants, which can be interpreted as hyperbolic version of GCN and GAT. *Our method*: GIL derives the geometry interaction learning to leverage both hyperbolic and Euclidean embeddings on graph topology.

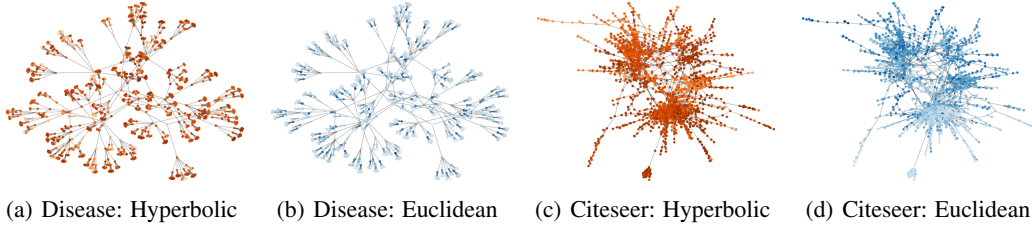

| (a) Disease: Hyperbolic | (b) Disease: Euclidean | (c) Citeseer: Hyperbolic | (d) Citeseer: Euclidean |

Figure 3: Visualization of probability weight in Disease and Citeseer datasets. The darker the node color is, the greater the weight is.

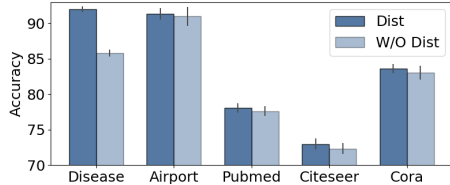

| Method | $\text{GIL}_{H \not\leftrightarrow E}$ | $\text{GIL}_{H \not\to E}$ | $\text{GIL}_{H \not\leftarrow E}$ | GIL |
|---|---|---|---|---|
| Disease | 84.25 | 88.58 | 88.98 | **90.70** |
| Airport | 87.02 | 87.37 | 89.91 | **91.33** |
| Pubmed | 78.18 | 78.23 | 78.31 | **78.91** |
| Citeseer | 70.80 | 72.06 | 71.26 | **72.97** |
| Cora | 80.50 | 83.15 | 82.20 | **83.64** |

Figure 4: Effect of distance attention in hyperbolic message propagation for NC in accuracy.

Table 3: Effect of different feature fusions for NC in accuracy.

Meanwhile, we construct the model EucHyp, which adopts the intuitive manner based on Eq. (20) to integrate features of Euclidean and hyperbolic geometries learned from GAT [24] and HGAT [21], seperately. For fair comparisons, we implement the Poincaré ball model with $c = 1$ for all hyperbolic baselines.

**Parameter Settings.** In our experiments, we closely follow the parameter settings in [19] and optimize hyperparameters on the same dataset split for all baselines. In node classification, we use the 30/10/60 percent splits for training, validation and test on Disease dataset, 70/15/15 percent splits for Airport, and standard splits in GCN [2] on citation network datasets. In link prediction, we use 85/5/10 percent edges splits on all datasets. All methods use the following training strategy, including the same random seeds for initialization, and the same early stopping on validation set with 100 patience epochs. We evaluated performance on the test set over 10 random parameter initializations. In addition, the same 16-dimension and hyper-parameter selection strategy are used for all baselines to ensure a fair comparison. Except for shallow methods with 0 layer, the optimal number of hidden layers for other methods is obtained by grid search in [1, 2, 3]. The optimal $L_2$ regularization with weight decay [1e-4, 5e-4, 1e-3] and dropout rate [0.0-0.6] are obtained by grid search for each method. Besides, different from the pre-training in HGCN, we use the same initial embeddings for node classification on citation datasets for a fair comparison. We implemented GIL using the Adam optimizer [31], and the geometric deep learning extension library provided by [32, 33].

## 4.2 Experimental Results

**Node Classification and Link Prediction.** In Table 2, we report averaged ROC AUC for link prediction, and for node classification, we report Accuracy on citation datasets and F1 score on rest two datasets, following the standard practice in related works [19]. Compared with baselines, the proposed GIL achieves the best performance on all five datasets in both tasks, demonstrating the comprehensive strength of GIL on modeling hyperbolic and Euclidean geometries. Hyperbolic methods perform better than Euclidean methods on the datasets where hyperbolic features are more significant, such as Disease and Airport datasets, and vice versa. GIL not only outperforms the hyperbolic baselines in the dataset with low hyperbolicity, but also outperforms the Euclidean baselines in the dataset with high hyperbolicity. Furthermore, compared with EucHyp which is a simple combination of embeddings from two spaces, GIL shows superb performance in every task, demonstrating the effectiveness of our geometry interaction learning design.

**Ablation.** To analyze the effect of interaction learning in GIL, we construct three variants by adjusting the feature fusions in Eq. (12) and Eq. (13). The variant with no feature fusion between two spaces, i.e. update above two equations as $h'_{\mathbb{D}_c} = h_{\mathbb{D}_c}$ and $h'_{\mathbb{R}} = h_{\mathbb{R}}$, is denoted as $\text{GIL}_{H \not\leftrightarrow E}$. If only the Euclidean features are fused into hyperbolic and the reverse fusion is removed, i.e. keep Eq. (12)

unchanged and update Eq. (13) as $\boldsymbol{h}'_{\mathbb{R}} = \boldsymbol{h}_{\mathbb{R}}$, we denote this variant as $\text{GIL}_{H \not\to E}$. Similarly, we have the third variant $\text{GIL}_{H \not\leftarrow E}$. Table 3 illustrates the contribution of interaction learning in node classification. As we can see, variants with feature fusion perform better than no feature fusion. Especially, the addition of dual feature fusion, e.g. GIL, provides the best performance. For the distance-aware attention proposed in hyperbolic message propagation, we construct an ablation study to validate its effectiveness in Figure 4. The model without the enhancement of distance attention not only decreases the overall classification accuracy on all datasets, but also increases the variance.

**Visualization.** To further illustrate the effect of probability assembling, we extract the maximum connected subgraph in Disease and Citeseer datasets and draw their topology in Figure 3, in which the shading of node color indicates the probability weight. We can observe that, on Disease dataset with hierarchical structure, the hyperbolic weight is greater than Euclidean overall, while on Citeseer dataset, the probability of both spaces plays an significant role. Moreover, the boundary nodes have more hyperbolic weight in general. It is consistent with our hypothesis that nodes located at larger curvature are inclined to place more trust on hyperbolic embeddings. The probability assembling provides interpretability for node representations to some extent.

# 5   Conclusion

This paper proposed the Geometry Interaction Learning (GIL) for graphs, a comprehensive geometric representation learning method that leverages both hyperbolic and Euclidean topological features. GIL derives a novel distance-aware propagation and interaction learning scheme for different geometries, and allocates diverse importance weight to different geometric embeddings for each node in an adaptive way. Our method achieves state-of-the-art performance across five benchmarks on node classification and link prediction tasks. Extensive theoretical derivation and experimental analysis validates the effectiveness of GIL method.

## Impact Satement

Graph neural network (GNN) is a new frontier in deep learning. GNNs are powerful tools to capture the complex inter-dependency between objects inside data. By advancing existing GNN approaches and providing flexibility for GNNs to capture different intrinsic features from both Euclidean spaces and hyperbolic spaces, our model can be used to model a wider range of complex graphs, ranging from social networks and traffic networks to protein interaction networks, for various applications such as social influence prediction, traffic flow prediction, and drug discovery.

One potential issue of our model, like many other GNNs, is that it provides limited explanation to its prediction. We advocate peer researchers to look into this to enhance the interpretability of modern GNN architectures, making GNNs applicable in more critical applications in medical domains.

## Acknowledgments and Disclosure of Funding

This work was supported by the NSFC (No.11688101 and 61872360), the ARC DECRA Project (No. DE200100964), and the Youth Innovation Promotion Association CAS (No.2017210).

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
