[Reviews · NeurIPS 2020]

Review 1

Summary and Contributions: This paper presents another graph embedding technique based on hyperbolic spaces and convolutions. The topic is interesting, but has been very widely investigated over the recent years. Contrary to the already rich literature on graph embeddings (as well as hyperbolic spaces), this paper utilizes jointly hyperbolic and euclidean embeddings. Basically, the proposed recipe seems to be the following: 1) Apply message passing in the hyperbolic and euclidean space(using a GAT -- an attention mechanism) 2) Transform the Euclidean features to hyperbolic space 3) Merge the features in an adaptive way.

Strengths: 1) It is the first work to try to leverage both the Euclidean and hyperbolic features. 2) It remains scalable. Although, it would have been interesting to see some running times -- the mobius multiplication, and operating in non-Euclidean space can be more costly than euclidean.

Weaknesses: 1) __Over explored topic, and ``okay`` results__ The topic (graph embedding, hyperbolic spaces) has been very well explored of late. As such, this paper suffers from the huge difficulty of making a case for yet another graph embedding technique: what justifies so many transformations and back and forth between spaces? The results that the authors show are very competitive and seem to beat SOA methods. While the accuracy seems higher, it is not necessarily statistically significant in many of the presented examples. It would have been interesting to make a stronger case for these results by adding running time, or other performance metrics. 2) __Few experiments:__ The authors present only two sets of experiments (on several datasets, but the experiment is the same): node classification and link prediction. It would have been interesting to try out other task, such as graph classification for instance. 3) Lack of code ######### Rebuttal Update #### After reviewing the authors' rebuttal and going through their submission, I have decided to increase the score of their submission. I have revised my opinion: while I maintain that this is an overly explored topic (and I wonder if the sample size necessary to obtain good results is reasonable enough to make the method practically usable in real life, to tackle medical applications for instance, as the authors mentioned), the authors' method does provide some improvement upon state-of-the-art method, and nicely extend the discussion in this particular area.

Correctness: No problem detected.

Clarity: Yes.

Relation to Prior Work: Yes.

Reproducibility: No

Additional Feedback:


Review 2

Summary and Contributions: This paper proposes GIL, a new type of graph neural networks (GNNs) that are suitable for graphs without an underlying constant curvature for which no computationally efficient Riemannian manifold could be used. The main idea is to leverage both the power of Euclidean and hyperbolic geometries and corresponding GNNs. The proposed model trains in parallel two GAT networks, one Euclidean and one hyperbolic, followed by cross interactions between their embeddings that is meant to encourage the interaction between the two geometries. Finally, the resulting embeddings are fed through Euclidean and hyperbolic softmax MLR layers for node and edge classification tasks. Carefully worked experiments on 5 datasets and 2 tasks show consistent and significant improvements w.r.t. the baselines. Ablation studies are also nicely done.

Strengths: See above.

Weaknesses: - My main concern is regarding the reproducibility of this method. This is a heavy black box model. Implementation details are almost entirely missing (even the number of layers is not specified). No code is currently published. Will the authors publish the full code and evaluation settings that would allow full reproducibility of these results ? I will be willing to increase my current rating if this issue would be addressed. - Claimed contribution in lines 77-78: the authors do not mention, cite or use as baselines any of the recent works on products of constant curvature spaces, e.g. [1,2,3]. Technically, these methods could jointly learn both Euclidean and hyperbolic representations. How does the proposed model compare against GNNs that embed graphs in products of constant curvature spaces with fixed or learnable curvatures, e.g. [2] ? - Line 32: "although they may suffer the problem of permutation variance " --> as far as I am aware, none of the hyperbolic GNN models suffers from this problem. Can the authors detail this statement? - line 119: another option is to use gyro-midpoints or Riemannian Frechet means, e.g. see [2,4]. - lines 158-160: "It is worth mentioning that after the feature fusion, the node embeddings [...] maintain properties and structures of the original space." -----> This reads like a vague statement to me. Can you please clarify why does it preserve those ? - line 252: "We evaluated all methods over the same random seed for 10 runs" --> why do you observe any differences if you use the same random seed ? [1] Learning mixed-curvature representations in product spaces, Gu et al, ICLR 2019 [2] Constant Curvature Graph Convolutional Networks, Bachmann et al, ICML 2020 [3] Mixed-curvature Variational Autoencoders, Skopek et al, ICLR 2020 [4] Differentiating through the Fr\'echet Mean, Lou et al, ICML 2020 ================================ LE (after rebuttal): the authors partially addressed my comments. I will increase my score. I am still not sure how this work compares to embeddings of graphs in products of constant curvature spaces that have an overall variable curvature.

Correctness: See above. This is largely a deep learning paper, there is no theory. The empirical methodology seems valid.

Clarity: Paper is clearly written.

Relation to Prior Work: Please see some missing citations above in the "Weaknesses" section.

Reproducibility: No

Additional Feedback: The current method is a heavy black box model. No code is currently published. Will the authors publish the full code and evaluation settings that would allow reproducibility of their method ?


Review 3

Summary and Contributions: The paper proposed a new method to address an important problem for geometric embedding. The authors argued that single geometric embedding lacks the flexibility to model complex geometries on graphs. To address this issue, this paper proposed a new method which can make an interaction learning from both Euclidean and hyperbolic geometries for graphs with complex geometries. The experimental validation demonstrates the significant improvement compared with baselines.

Strengths: 1. Motivation. This paper presents an important problem on geometry interaction learning on graphs. The existing works only consider single geometry embedding, such as Euclidean or hyperbolic, which lack the flexibility to model complex geometries and cannot fully capture the geometric information available in graphs. While this work tries to capture both geometric features simultaneously in a graph, and gives each node the power to decide which geometric features is more reliable for the downstream tasks. 2. Novelty. The authors provide a novel framework that unifies Euclidean and hyperbolic geometries, two previously disparate research areas, utilizing the strength of both two geometries and maintaining the geometric properties of each space, directly from two essential parts, interaction learning and probability assembling. It is a significant contribution for moving towards a more general setting in graph geometry learning, without knowing any prior structural information. 3. Soundness. The extensive theoretical analysis and empirical evaluations support major claims in the paper, with SOTA results and convincing experimental analysis. And the further visualization also shows the interpretability of the learnable embeddings and different weights. 4. Relevance. There has been an increased interest in geometric deep learning on graphs. The significant and contributions of this paper are undisputed on this domain.

Weaknesses: Overall, I found the paper interesting and practically useful; although I believe some additions to the empirical evaluation can improve the impact of the paper if there is any space left, such as the low-dimensional modeling ability of the proposed method, along with some experimental details, including the search space of hyper-parameters and the detailed description of the compared methods.

Correctness: The technical content of the paper is correct and major claims are well supported by the theoretical analysis and empirical evaluations. There is an unclear part for the optimizer. Because there are several parameters involving both Euclidean and hyperbolic geometries in this paper, such as Euclidean parameters W in Eq. (8), a in Eq. (10), \beta in Eq. (12), and hyperbolic parameters b in Eq. (8). As shown in the experimental settings, the Adam optimizer is selected to train the model. I think that the authors transfer the hyperbolic parameters to Euclidean parameters as the trainable objects to use the optimizer uniformly. This part should be clarified for correctness.

Clarity: Yes. This paper is well organized and easy to follow. The notations are consistent throughout the document, with necessary description clearly stated. The model structure depicted in Figure (2) gives a clear and intuitive picture of the overall framework.

Relation to Prior Work: Yes. Authors provided the relevant related works and elegantly show how their work improves on the existing literature in Introduction section. The existing works only consider single geometry embedding, either Euclidean or hyperbolic, while this work presents an interaction learning from both Euclidean and hyperbolic geometries for modeling graphs, which is a brand-new attempt in geometric deep learning.

Reproducibility: Yes

Additional Feedback: In reality, many complex graphs may exhibit more than one geometric structure, and we cannot know in advance which geometric features are more appropriate for the nodes in graph representation learning. And the paper introduces this significant problem and then proposes a comprehensive geometry interaction learning method DIL in response to the problem. In the DIL, the different geometric features can be adaptively updated in local space and then conformally adjusted from dual space, which is not a trivial attempt between two different geometrics with independent close-form operations. Meanwhile, the intuitive interaction manner EucHyp is not ideal as the results shown in Table 2. In my view, this is an effective attempt to address a meaningful problem. The paper is generally well-written and structured clearly. However, there are still some issues that need to be clarified. 1)The values of hyperbolicity distribution are supposed to denote the proportion of corresponding hyperbolicity, which should be stated clearly. 2)For the experimental settings, the authors basically follow the same settings in related works. However, it would be beneficial for readers to compare and verify if details of settings could be attached to the additional material, such as search space of hyperparameters.


Review 4

Summary and Contributions: This paper proposes a graph geometry interaction learning method which allows to embed a graph into a Euclidean space and a hyperbolic space simultaneously. More specifically, mapping between two spaces, distance-aware attention mechanism for hyperbolic propogation, and a feature interaction learning module are mainly explored. Experiments around node classification and link prediction shows promising results.

Strengths: The theoretical grounding such as Hyperbolic geometry, graph attention networks and extended improved study such as geometry message propagation, probability assembling are clear and reliable. The visualization of Euclidean and hyperbolic embeddings is intuitive. The topic is around improving GNN to extract more features and is relevant to NeurIPS community.

Weaknesses: Explanation around why applying exponential and logarithmic mapping features and feature interaction can be proved in detailed. The model can be seen as a combination of an Euclidean and a hyperbolic model. Novelty is limited. Also, the model performs an online ensemble of these two models. What if the two models are trained offline, before being combined by a weight hyperparameter?

Correctness: Yes.

Clarity: Yes.

Relation to Prior Work: Yes, the proposed interaction learning aims to collect the features from hyperbolic and Euclidean spaces, and discussion about prior works are well organized and clarified.

Reproducibility: Yes

Additional Feedback: The authors address my concerns during the rebuttal period. The scores are maintained the same.

[Author Response · NeurIPS 2020]

We thank the reviewers for their insightful and positive feedback. We are encouraged they found our motivation
and idea to be clear (R4), new (R2, R3), and novel (R3), well organized and clarified w.r.t. prior works (R3, R4),
and our theoretical analysis reliable/well-supported (R3, R4). We are glad they found our method to be intuitive
(R3, R4), evaluated with extensive/convincing/carefully worked experiments (R2, R3), and achieving consistent
significant/promising improvements (R2, R3, R4). We address reviewers comments below and will incorporate all
feedback in the final version.

Q. [R1] "Over explored topic, and "okay" results."
A. We disagree with this statement. (1) The geometric learning on graph has attracted an increasing attention, because it
is an important topic; but still it has many unexplored questions such as interaction learning, where this paper tries to
work out. (2) Our method achieves SOTA results w.r.t. baselines, evidenced by averaged performance with standard
deviation, and the further ablation study and visualization show significant improvements, which are endorsed by other
three reviewers.

Q. [R1]."What justifies so many transformations and back and forth between spaces?"
A. In order to bridge the geometry gap for interactive learning and maintain conformal invariance of each space, we
utilize the *exponential* and *logarithmic* mappings to transform features between spaces.

Q. [R1] "It would ... interesting to make a stronger case ... by adding running time, or other performance metrics"
A. The time complexity is analyzed quantitatively in Section 3.3, which is more exact and computationally efficient.
The comment about the time consumption of basic operations in different spaces is reasonable, where operations in
hyperbolic space indeed consume more time than Euclidean space. For instance, GAT and HGAT cost 3.48s and 10.59s
respectively on Cora in node classification. The extra time consumed is also within an acceptable range and doesn't
affect it's extendibility.

Q. [R1] "Few experiments, it would be interesting to try out other task, such as graph classification."
A. We disagree with "few experiments". Convincing/extensive experiments are endorsed by other reviewers. Besides,
our approach is to endow each node the freedom to determine the importance of each geometry space, therefore, the
node-level tasks are sufficient to validate the effectiveness of GIL, including node classification and link prediction.

Q. [R1] "Lack of code." [R2] "Will the authors publish the full code and evaluation settings ... ?"
A. As we cannot report an external link here during rebuttal, we will open the source code after blind review. For
evaluation settings, we closely follow the parameter settings in HGCN mentioned in Section 4.1. The detailed settings
are supplemented as follow. Model configuration: Except for shallow methods with 0 layer, other methods use the
same 2 hidden layers. Training configuration: All methods use the following training strategy, including the same
random seeds for initialization, and the same early stopping on validation set with 100 patience epochs. We measure
performance on the test set over 10 random parameter initializations. The optimal $L_2$ regularization with weight decay
[1e-4, 5e-4, 1e-3] and dropout rate [0.0-0.6] are obtained by grid search for each method. We will add the details and
release our source code in the final version.

Q. [R2] "How does the model compare against ... constant curvature spaces with fixed or learnable curvatures, e.g.[2]?"
A. The constant curvature spaces are a unified form of different curvature spaces. The difference between proposed
method and the latest works about constant curvature spaces, is that these methods still embed the entire graph in a
space of uniform curvature, although the curvature is fixed or learnable. While our method give each node privilege
to determine proper features from different curvature spaces, specifically using hyperbolic and Euclidean space here.
More importantly, there is no interaction learning between different spaces in existing works. Thanks for pointing the
newly related work though it was published after NeurIPS submission deadline. We will add more discussion in the
final version.

Q. [R2] ...the problem of permutation variance –>"...Can the authors detail this statement?"
A. Hyperbolic GNNs suffer from the problem of permutation invariance, because the basic operations in hyperbolic
space such as addition cannot hold the property of commutative and associative, it cannot maintain the permutation
invariance if apply GNN operate directly in hyperbolic space. Therefore, transfer operations such as aggregation into
tangent space is a intrinsic choice to deal with this problem.

Q. [R2] "why does it preserve properties and structures of the original space?"
A. Because of different closed-form operations in each space, the feature interaction learning first transform different
features into its own space and then fuse them, therefore, it can maintain conformal invariance of each space.

Q. [R4] "What if the two models are trained offline, before being combined by a weight hyperparameter?"
A. The suggested method was added as a baseline EucHyp in the paper. The results show that our algorithm significantly
outperform this method in Table 2.

[Meta-Review · NeurIPS 2020]

The paper was reviewed by 4 expert reviewers who appreciated the paper and the rebuttal. The updated reviews contain further advice for the camera ready version, and the authors should consider this.